# Targeted Diode Laser Therapy for Oral and Perioral Capillary-Venous Malformation in Pediatric Patients: A Prospective Study

**DOI:** 10.3390/children10040611

**Published:** 2023-03-24

**Authors:** Angela Tempesta, Fabio Dell’Olio, Rosaria Arianna Siciliani, Gianfranco Favia, Saverio Capodiferro, Luisa Limongelli

**Affiliations:** Complex Operating Unit of Odontostomatology, Department of Interdisciplinary Medicine, Aldo Moro University, 70121 Bari, Italy; angelatempesta1989@gmail.com (A.T.); f.dellolio.odo@outlook.it (F.D.); gianfranco.favia@uniba.it (G.F.); saverio.capodiferro@uniba.it (S.C.); luisanna.limongelli@gmail.com (L.L.)

**Keywords:** diode laser, photocoagulation, pediatric patients, pediatric dentistry, vascular anomalies, vascular malformations, intraoral ultrasonography, ultrasound, head and neck, oral

## Abstract

Background: This study describes the management protocol for capillary-venous malformations in pediatric patients and reports the epidemiology of diagnosed and treated cases at the Unit of Odontostomatology of the Aldo Moro University of Bari from 2014 to 2022. Methods: The authors classified the intraoral and perioral capillary-venous malformations by superficial diameter (<1 cm, 1–3 cm, >3 cm) and ultrasonographical depth extension (≤5 mm, >5 mm). All patients underwent pulsed-mode diode laser transmucosal photocoagulation (8–12 W/cm^2^); those with malformations that were wide (>3 cm) and deep (>5 mm) received intralesional photocoagulation, too (13 W/cm^2^). The children received general anesthesia based on their compliance and lesions’ extension. The follow-up lasted six months. Results: A total of 22 females and 14 males (age range 4–18 years) presented 63 capillary-venous malformations. Five patients with Sturge–Weber syndrome, seven with hereditary hemorrhagic telangiectasia, and five with angiomatosis showed multiple malformations. The authors found no intraoperative or postoperative complications. Seventeen patients with lesions >1 cm and >5 mm deep required multiple laser sessions to heal. Conclusion: The results of the current study support diode laser photocoagulation as the gold standard for the treatment of intraoral and perioral capillary-venous malformations in pediatric patients.

## 1. Introduction

Vascular anomalies (VAs) affect any area of the human body, and the head and neck region accounts for about 60% of those diagnosed in children, showing an incidence of 1:22 pediatric patients [1]. VAs require prompt diagnosis and correct management in pediatric patients because of the risks and the long-term functional and psychological consequences depending on the number, size, localization, and extension of lesions [2]. In 2018, the International Society for the Study of Vascular Anomalies (ISSVA) released the last update of its classification system for VAs, which distinguished vascular tumors (VTs) and vascular malformations (VMs) by considering the different clinical–histological characteristics [3,4]. VTs arise from the neoplastic endothelial cells and often undergo involution, whereas VMs occur because of local dysregulation of pathways of embryogenesis and angiogenesis [3,4]. The ISSVA classification further distinguishes high-flow and slow-flow VMs by considering the flow intensity and recognizes venous, lymphatic, capillary, arteriovenous, and combined malformations by studying the types of vessels in the lesions [3,4]. Eventually, VMs can occur as angiomatoses and as signs of syndromes such as Sturge–Weber (SWS) and hereditary hemorrhagic telangiectasia (HHT) developing in childhood. Intraoral and perioral VMs require a multidisciplinary approach, long-term treatment plan, and prolonged follow-up [5]. Other than clinical examination, intraoral high-definition ultrasonography (US) is the first imaging technique for the diagnosis of intraoral and perioral VMs because it is non-invasive and inexpensive [6,7]. B-Mode US shows VMs as homogenous hypoechoic lesions with irregular margins associated with phleboliths in up to 16% of cases. In addition, color and power Doppler distinguish high-flow and low-flow malformations [1]. Magnetic resonance imaging (MRI), both with and without the administration of contrast, is the method used to complete the study of large intraoral and perioral VMs by defining the site, relationships with adjacent structures, and internal architecture of the lesions. VMs usually develop medium- or low-intensity signals in MRI and appear hyperintense only in the case of intralesional fat. Infants, preschool children, and even older pediatric patients often undergo MRI under general anesthesia because such a technique requires a high grade of compliance, whereas the same patients easily accept intraoral US. Clinicians must consider the advantages and limitations of US and MRI in children because those techniques are useful for the initial diagnosis of intraoral and perioral VMs as well as for follow-up [8]. The indication of surgical treatment is the removal of small intraoral capillary venous malformations (CVMs) with a low risk of recurrence and bleeding, whereas the excision of perioral VMs is not indicated in pediatric patients because of the risk of developing inesthetic and functionally impairing scars [9,10]. Intralesional sclerotherapy—consisting of the injection of sclerosing agents to induce inflammation and thrombosis within the malformation—and many sclerosing agents have been proposed, each with specific characteristics, indications, and risks for the treatment of intraoral and perioral arteriovenous VMs [11]. In addition, embolization is useful for occluding the vessels of arteriovenous malformations localized in the maxillo-facial region and uses permanent or temporary embolizing agents [12]. Several researchers studied the treatment of complicated intraoral and perioral VMs by administration of immunosuppressant agents such as mTOR inhibitors (e.g., sirolimus, rapamycin) but achieved contrasting results [13]. Nowadays, coagulative laser therapy is the gold standard used for intraoral and perioral CVMs in pediatric patients because of its low- or non-invasiveness, the chance to treat children under local anesthesia with or without conscious sedation, and the availability of devices targeting hemoglobin. The main laser techniques for CVMs are transmucosal thermo-photocoagulation (TMT) and intralesional photocoagulation (ILP). TMT is a no-contact application, whereas ILP involves the insertion of the fiber—bare or through an intravascular catheter—into the lesion and is the treatment of choice for large and deep intraoral and perioral CVMs [6]. When the type, localization, and size of the lesions prevent complete removal, the treatment aims to improve the patient’s quality of life by relieving symptoms, reducing thrombotic risk, and solving esthetic issues [14]. The current study describes the protocol for the diagnosis and laser treatment of CVMs in pediatric patients of the Unit of Odontostomatology of the Aldo Moro University of Bari and reports epidemiological data for cases treated from 2014 to 2022.

## 2. Materials and Methods

The authors conducted the current study according to the principles of the Declaration of Helsinki, and the Independent Ethical Committee of the Aldo Moro University of Bari approved the study protocol (study number 4576, code 1443/CE). The authors included pediatric patients (age range 0–18 years old) with a clinical diagnosis of intraoral and/or perioral CVMs diagnosed and treated at the Unit of Odontostomatology of the Aldo Moro University of Bari from 2014 to 2022. The parents of the included patients provided informed consent for the treatment plan and the use of the patients’ data for scientific research purposes. During the first visit, the authors recorded each patient’s medical history and clinical data of the CVMs to identify any syndromes associated with VMs, comorbidities, and pharmacological therapies in progress. The authors also gathered data regarding the number, location, tendency to bleed, and pain of the CVMs. The parents answered questionnaires aiming to investigate the functional, esthetic, or functional–esthetic impairments associated with the lesions. The authors classified the problems as functional if limited to disruption of eating, swallowing, and phonation. Such data were useful to establish the need and urgency of treatment. The authors classified the lesions based on their superficial diameter according to their previously published system into three groups, namely: group A for lesions <1 cm, B for lesions in the range of 1–3 cm, and C for lesions >3 cm [6]. The patients carrying B and C lesions underwent high-definition intraoral ultrasonography to measure the depth extension of the malformation. The authors performed the ultrasonographical examination by using an ultrasound system equipped with the linear “Hockey Stick” probe delivering a frequency of 18 MHz (“Logic 9”, General Electric Healthcare, Chicago, IL, USA) to study the intraoral CVMs [6,7]. In addition, the authors studied the perioral cutaneous CVMs by using a convex-type US probe delivering 5 MHz ultrasounds. The use of Doppler also allowed the assessment of the speed of the flow to distinguish high- and slow-flow lesions. According to the findings of the US, the lesions were further classified according to their depth into B1 and C1 groups if the depth was ≤5 mm and B2 and C2 groups for those with a depth >5 mm [6]. Only children with lesions of group B2 and/or C2 underwent MRI. The photocoagulation protocol was carried out using a GaA1As-A2G diode laser (“Surgery35”, AsG srl, Turin, Italy) delivering light of 800 ± 10 nm mean wavelength by a flexible quartz fiber of 320 µm in diameter (Table 1). The authors performed TMT in all CVMs by using a power output between 8 and 12 W/cm^2^, pulsed mode (t-on = 190 ms, t-off = 250 ms), and placing the tip of the fiber at 2–3 mm above the surface of the target lesion. In addition to the TMT, the authors treated all B2 and C2 lesions by using ILP, introducing the bare fiber 5 mm into the malformations, and providing a power output of 13 W/cm^2^ in pulsed mode [6]. This photocoagulation protocol required the application of ice packs during and after the laser session to prevent the onset of thermal damage to the target tissues.

At the end of the session, the lesions turned from a dark red-bluish color to grey or white as a sign of photocoagulation. The patients with multiple and/or larger lesions underwent at least two laser sessions complying with a healing period of five weeks after each photocoagulation. Each laser session required the infiltration of local anesthetic without vasoconstrictor (mepivacaine 3%) to avoid reducing the blood supply to the target tissue. The children underwent laser photocoagulation of the symptomatic CVMs under general anesthesia, conscious sedation, or local anesthesia alone, according to the degree of cooperation, which corresponded to their age and the presence of cognitive deficits. During the postoperative period, all patients received the application of a gel based on hyaluronic acid and amino acids (Aminogam^®^ gel) to the site of photocoagulation to accelerate wound healing [6]. All patients underwent clinical follow-up visits on the 1st, 3rd, 5th, 7th, and 30th postoperative days and after three and six months. During each visit, the parents of the patients also answered a questionnaire to collect data on postoperative pain experienced by the children.

## 3. Results

Table 2 shows the characteristics of the 36 patients that matched the inclusion criteria of the study, 22 females (61.11%) and 14 males (38.89%) with a mean age of 14 years (range 4–18 years). A total of 24 patients were without comorbidities and showing non-syndromic CVMs (66.67%), whereas medical histories revealed the diagnosis of SWS in 5 patients (13.89%) and HHT in 7 patients (19.44%). In addition, the HHT patients showed familial inheritance, which was maternal in five cases and paternal in the other two. Since 20 patients showed a single CVM (55.56%) and 16 carried multiple malformations (44.44%), the authors treated 63 lesions. Among the syndromic children, only one HHT patient showed a single CVM (8.33%, HHT), in contrast to the remaining eleven cases who carried multiple malformations (91.67%). The five non-syndromic patients with multiple CVMs were classified as angiomatoses (31.25%). The diagnosis was telangiectasia for 19 lesions and slow-flow CVMs for the remaining 44. The localization of the lesions was the tongue for 14 lesions (22.22%), palate for 6 lesions (9.52%), lips for 7 lesions (11.11%), gingiva/vestibular fornix for 9 lesions (14.29%), buccal mucosa for 16 lesions (25.40%), and perioral skin for 11 lesions (17.46%). According to the superficial diameter, 23 lesions belonged to the A group (36.51%), 18 to the B group (28.57%), and 22 to the C group (34.92%). Patients affected by non-syndromic angiomatosis and SWS accounted for 18 lesions of the C group (81.81%). After the intraoral US and MRI study of the depth of the lesions, eight CVMs were classified as in the B1 group (44.44%) whereas ten were classified as in the B2 group (55.56%). In addition, the authors further classified nine CVMs into the C1 group (40.90%) and thirteen into the C2 group (59.10%).

The 11 CVMs that involved the perioral skin caused esthetic issues (17.46%), the 8 lesions that occurred in the lips caused functional-esthetic issues (12.70%), and the 44 intraoral malformations caused only functional problems (69.84%). The children did not experience pain associated with the CVMs, but the parents reported that 12 lesions bled after incidental trauma (19.05%). Only one patient with a C2-group CVM underwent general anesthesia to receive photocoagulation (2.78%), whereas the authors treated 22 patients under conscious sedation by benzodiazepines (61.11%). The remaining 13 patients received the photocoagulation under local anesthesia alone (36.11%). The authors did not experience intraoperative complications such as carbonization of the target site or bleeding due to the breaking of malformations. During the postoperative period, the patients reported mild pain manageable with the administration of acetaminophen. In addition, the treated sites did not show bleeding, scarring, ulcers, necrosis, or infection. After the first session of photocoagulation, 19 patients showed complete healing (52.78%), whereas 17 patients (47.22%) required multiple sessions because of CVMs belonging to the C groups and/or B2 group. Nine patients required two sessions to heal (25.00%, e.g., Figure 1) and eight required three sessions (22.22%). Among the 63 treated lesions, 58 (92.06%) healed without recurrence after a follow-up period of 6 months. The other five lesions (7.94%) improved significantly but required further sessions out of the study period (e.g., Figure 2).

## 4. Discussion

From 50% to 83% of pediatric VAs occur in the head and neck region with an incidence ratio of 1:22 [1,2]. Therefore, dentists are important for the diagnosis and management of pediatric VAs [15]. In addition, these lesions have functional, esthetic, and psychological implications for pediatric patients and their families. In the current study, CVMs caused esthetic issues in 17.46% of cases and had functional–esthetic implications in 12.70%, which also carried psychological consequences for the children [2]. The diagnosis of VMs is still a challenge because of the many lesions with similar clinical appearances. Venous malformations are slow-flow vascular diseases that occur as single lesions in 90% of cases and show a tendency to grow without spontaneous regression [2,8]. Superficial venous malformations appear bluish and show a soft consistency. Mutations of the TEK and PIK3CA genes are frequent in those lesions [8]. Venous malformations appear in histological exams as mitotically inactive, large, and irregular channels delimited by flat endothelium surrounded by smooth muscle cells. Capillary malformations are congenital single, multiple, or confluent dark-red lesions (namely, “port-wine stains”) with a variable diameter, appearing several days after birth. In addition, capillary malformations grow together with the body, becoming thick and cobbled [12]. The histology of those lesions shows abnormal dilated capillary-like vessels with a thin endothelium and without surrounding smooth muscle cells. Angiomatosis is a benign non-syndromic form of multiple CVMs affecting large body segments, and most cases develop during the first two decades of life [16,17]. Angiomatoses of the oral and perioral areas occur as spots or as plaque-like or nodular lesions that are red in the skin and bluish-red in the mucosa [16,17]. SWS is a rare congenital disorder that affects 1:50000 births and causes cutaneous, neurological, and ocular manifestations [18]. The cutaneous manifestations arise as congenital unilateral VMs of the face with a distribution following one of the three branches of the trigeminal nerve. The leptomeningeal VMs are the main neurological manifestations and induce the formation of cerebral cortex calcium deposits, which cause epileptic seizures, mental retardation, hemiplegia, and hemiparesis. Glaucoma, buphthalmia, hemianopsia, and choroidal hemangioma are ocular manifestations of this syndrome [19]. Eventually, SWS appears as intraoral CVMs, causing macrocheilia, gingival hyperplasia, macroglossia, asymmetries, and malocclusions [20]. SWS patients carry mutations of the RAS p21 protein activator 1 gene, which contributes to cell proliferation [20]. HHT is a rare disease that causes systemic fibrovascular endothelial dysplasia developing telangiectasias and aneurysms as VMs [21]. The prevalence of HHT is about 1:5000–1:8000 cases and is an autosomal dominant disease, although 20% of cases occur because of spontaneous mutations. Among the five types of HHT, the most common are type I and II, which occur because of mutation in the ENG gene (9q34.1) and the ALK1 gene (12q11-q14), respectively [22]. The typical clinical manifestations of HHT develop throughout life until the mean age of 40 years. The multiple mucocutaneous telangiectasias are representative of HHT and involve the nasal mucosa (68–100%) and the oral cavity (58–79%). Arteriovenous malformations occur in the gastrointestinal tract, liver, brain, and spinal cord [23]. The histology of the telangiectasias shows dilated vessels with a thin layer of smooth muscle and rare elastic fibers, which is the reason for the typical fragility and high risk of hemorrhage. The most frequent symptom is recurrent and spontaneous epistaxis, which can cause iron deficiency anemia [22]. The intraoral and perioral CVMs require an early diagnosis, a multidisciplinary approach, and a prolonged follow-up. During the diagnostic phase, the clinical examination assesses the characteristics of the CVMs, the growth pattern, the age of onset, the presence of secondary localizations, and the functional–esthetic and psychological implications, to determine the need for treatment and the type of treatment [2]. The ISSVA system is the current reference classification for studying CVMs [3,4]. The authors integrated the ISSVA classification with the tridimensional staging system introduced by Limongelli et al., which distinguishes CVMs according to the superficial diameter (<1 cm, in the range 1–3 cm, and >3 cm) and the depth (≤5 mm or >5 mm) and allows the set-up of a targeted photocoagulation technique for each lesion [6]. In the current study, the use of HD intraoral US provided data about the depth of the lesions, limiting the cases of children that underwent general anesthesia for MRI to compensate for the lack of compliance [8]. In the current study, color and power Doppler HD intraoral US was also the elected imaging technique for the diagnosis of multiple intraoral and perioral CVMs with a diameter <1 cm, which would not be depicted by MRI [1,6,7,23]. In addition, as the ISSVA classification highlights, the presence of multiple syndromic CVMs implies the need for more than one treatment session and close follow-up because the syndrome causes the appearance of new primary lesions as manifestations of abnormal angiogenesis [3,4,19,21,22,23]. Voluminous CVMs require a long-term treatment plan, too [5]. In the current study, the parents filled out questionnaires to assess the functional and/or esthetic implications of the CVMs carried by their children, and the authors considered those findings when setting up the therapy. In addition, the authors opted for the photocoagulation of the CVMs because of the effectiveness and minimal invasiveness of this kind of treatment, which is useful to preserve the compliance of pediatric patients. Among all the lasers acting on the hemoglobin chromophore (e.g., Neodymium YAG, KTP), the diode laser is the one studied in the literature for the photocoagulation of intraoral and perioral CVMs as well as for the excision of benign and malignant neoformations of the oral cavity [16]. Diode laser induces photocoagulation of the lesions and consequent healing without adverse events such as intraoperative bleeding, postoperative bleeding, recurrences, and scars, which are the main risks associated with the treatment of those malformations [6,16,21,23]. The findings of the current study agree with the literature because all sites healed without severe adverse events, scars, or recurrences. The low invasiveness of the diode laser photocoagulation limited the use of general anesthesia to a case whose difficulty derived from the wide extension of the CVM and the low compliance of the patient due to age. Literature suggests surgical excision, sclerotherapy, and embolization as alternatives to laser photocoagulation. The surgical excision of extensive VMs is a challenge because few of those lesions show well-defined margins; therefore, recurrences, bleeding, and damage to adjacent anatomical structures are common [9,14]. Other authors support sclerotherapy as the gold-standard treatment for VMs because such a technique is minimally invasive and carries a low risk of intraoperative and postoperative bleeding and scarring formation [14,24]. Around 25% of cases show pain, edema, and wound infection, up to 30% develop necrosis and ulceration of the skin and soft tissues, and up to 5% of patients experience severe side effects such as anaphylaxis, pulmonary embolism, and cardiovascular collapse [8,14,24]. Embolization is an aid in the management of vascular malformations, especially those with high flow; however, embolization requires angiography to perform the procedure safely, and adverse effects such as pain, edema, skin and mucous ulcerations, neurological damage, anaphylaxis, and cardiovascular collapse are frequent. The current literature reports the use of immunosuppressors such as sirolimus as a new option for the treatment of large pediatric VMs that have not responded to other types of treatment. To date, the results of several studies differ, with some confirming the effectiveness of sirolimus and others showing the contrary [13]. In contrast with the techniques described above, in this study, no adverse events occurred when using laser photocoagulation. The operators need accurate knowledge of the characteristics of the laser to use appropriate powers and exposure times. The current study showed the management protocol for pediatric cases of CVMs developed by a tertiary referral hospital. The study protocol supported diode laser photocoagulation as a minimally invasive approach to treat single and multiple syndromic malformations in a cohort of children, even those affected by rare diseases. In addition, this management limited the need for general anesthesia in diagnostic and therapeutic phases. The main limitations of the study related to the short follow-up period.

## 5. Conclusions

The results of the current study support diode laser photocoagulation as the gold standard for the treatment of intraoral and perioral CVMs in pediatric patients. The management of those malformations requires deep knowledge of the lesions, adequate training in the use of laser devices, and consideration of the special needs of pediatric patients.

## Figures and Tables

**Figure 1 children-10-00611-f001:**
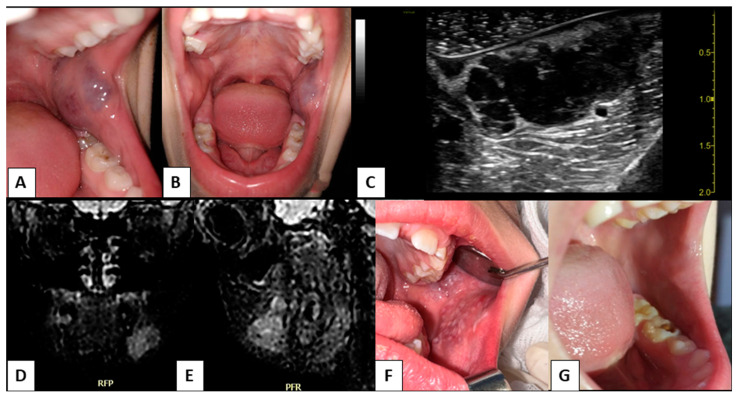
(**A**,**B**): capillary-venous malformation of left cheek in a six-year-old patient; (**C**): preoperative intraoral high-definition ultrasonography; (**D**,**E**): preoperative magnetic resonance imaging; (**F**): the lesion after transmucosal and intralesional thermo-photocoagulation; (**G**): follow-up after second session of laser photocoagulation.

**Figure 2 children-10-00611-f002:**
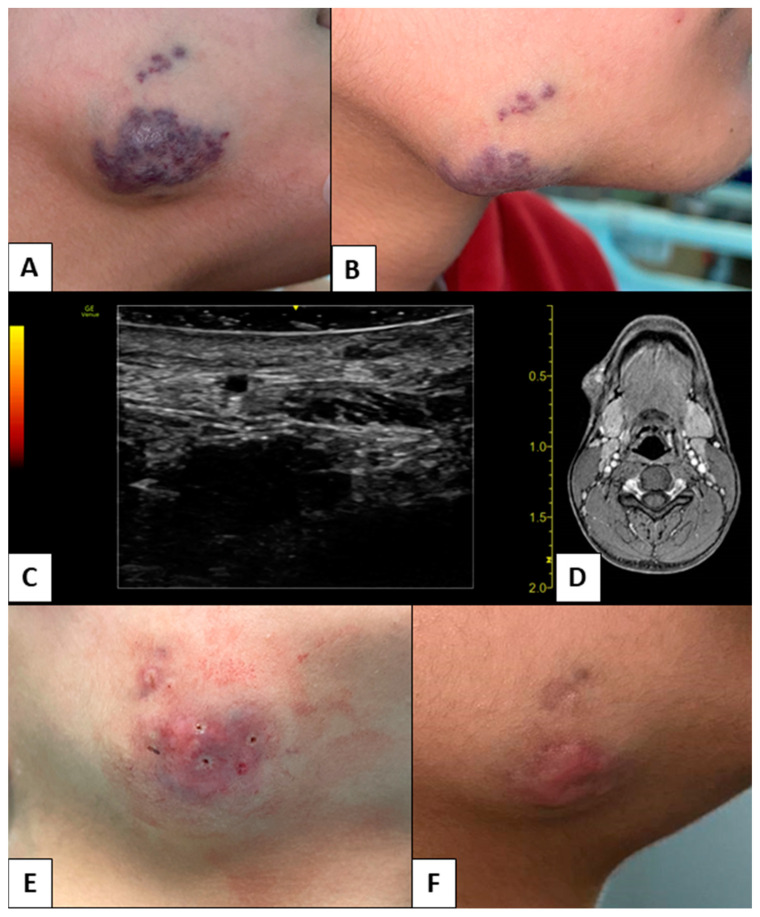
(**A**,**B**): a perioral capillary-venous malformation in a sixteen-year-old patient; (**C**): preoperative high-definition intraoral ultrasonography; (**D**): preoperative magnetic resonance imaging; (**E**): the lesion after transmucosal thermo-photocoagulation and intralesional photocoagulation; (**F**): The follow-up after the third laser session. The treatment is still in progress.

**Table 1 children-10-00611-t001:** Laser Parameters.

Parameters	Settings
Type of Laser	GaA1As-A2G Diode
Mean Wavelength (nm)	800 ± 10 nm
Type and Gauge (µm) of the Fiber	Quartz; 320 µm
Power Output for TMT (W/cm^2^)	8–12 W/cm^2^
Power Output for ILP (W/cm^2^)	13 W/cm^2^
Modality	Pulsed Mode
t-on (ms)	190 ms
t-off (ms)	250 ms

Abbreviations: TMT, transmucosal thermo-photocoagulation; ILP, intralesional photocoagulation.

**Table 2 children-10-00611-t002:** Characteristics of the Study Patients.

Characteristic	Data
*n*	36
Male/female	14/22
Mean age and age range	14 years; range 4–18 years
Syndromes (*n*)	SWS, 5 patients; HHT, 7 patients
Angiomatoses (*n*)	5 patients
Patients without comorbidities (*n*)	24 patients
Number of CVM	63
Group A CVM (*n*; %)	23 (36.51%)
Group B1 CVM (*n*; %)	8 (12.70%)
Group B2 CVM (*n*; %)	10 (15.87%)
Group C1 CVM (*n*; %)	9 (14.29%)
Group C2 CVM (*n*; %)	13 (20.63%)

Abbreviations: SWS, Sturge–Weber syndrome; HHT, hereditary hemorrhagic telangiectasia; CVM, capillary-venous malformations.

## Data Availability

The data presented in this study are available on request from the corresponding author. The data are not publicly available due to privacy and ethical restrictions.

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
