# Peer review of "Targeted Diode Laser Therapy for Oral and Perioral Capillary-Venous Malformation in Pediatric Patients: A Prospective Study"

_children, 2023, doi:10.3390/children10040611_

Round 1

Reviewer 1 Report

The article is well written with well described methods, results and discussion. however, there is one main concern. The title says it is a retrospective study which is a little misleading. Throughout the methodology it appears that the authors of this study carried out all the treatments and the publishing of this data was planned before all the treatments took place as questionaires were also used during before and after treatment. In this case it appears more likely to a prospective study then a retrospective study. Hence, it is important to modify the writing style of the methods or correct the title. In case it is a prospective study, please ensure to follow the strobe checklist guidelines or any similar guidelines that were followed.

Author Response

The authors thank Reviewer 1 for appreciating the manuscript and for providing such a precious suggestion. The manuscript has been revised as a prospective study.

Reviewer 2 Report

-I strongly suggest that data is presented in tables and the laser parameters are summarized in a table.

-The discussion is wordy and can be revised

Author Response

The authors thank Reviewer 2 for providing the current suggestions. The tables reporting the laser parameters and the characteristics of the patients have been added to the revised version of the manuscript. Anyway, the authors did not abbreviate the discussion section because the editors asked to make the manuscript longer than it was in the first version.

Reviewer 3 Report

The paper is interesting but it should not be classified as an article, but rather as a mixture of case series and narrative review. 

I would mention the limitation of the report, which is the short follow-up period of the patients, which makes it impossible to observe the recurrence rate. I would also add a table with a detailed description of the diseases that the patients suffered from. At the moment, only the abstract contains this information.

Author Response

The authors thank Reviewer 3 for the precious contribution to the improvement of the manuscript. The authors reported the limitation in the discussion section and added a table showing the description of the comorbidities and other characteristics of the study patients.